# Selective Synthesis of Benzimidazoles from *o*-Phenylenediamine and Aldehydes Promoted by Supported Gold Nanoparticles

**DOI:** 10.3390/nano10122405

**Published:** 2020-12-01

**Authors:** Marina A. Tzani, Catherine Gabriel, Ioannis N. Lykakis

**Affiliations:** 1Department of Chemistry, Aristotle University of Thessaloniki, University Campus, 54124 Thessaloniki, Greece; marina_tzani@hotmail.com; 2HERACLES Research Center, KEDEK, Laboratory of Environmental Engineering (EnvE-Lab), Department of Chemical Engineering, AUTH, 54124 Thessaloniki, Greece; katerinagabriel79@gmail.com

**Keywords:** gold nanoparticles, benzimidazoles, cyclization reaction, heterogeneous catalysis, *o*-phenylenediamine, thiabendazole

## Abstract

We investigated the catalytic efficacy of supported gold nanoparticles (AuNPs) towards the selective reaction between *o*-phenylenediamine and aldehydes that yields 2-substituted benzimidazoles. Among several supported gold nanoparticle platforms, the Au/TiO_2_ provides a series of 2-aryl and 2-alkyl substituted benzimidazoles at ambient conditions, in the absence of additives and in high yields, using the mixture CHCl_3_:MeOH in ratio 3:1 as the reaction solvent. Among the AuNPs catalysts used herein, the Au/TiO_2_ containing small-size nanoparticles is found to be the most active towards the present catalytic methodology. The Au/TiO_2_ can be recovered and reused at least five times without a significant loss of its catalytic efficacy. The present catalytic synthetic protocol applies to a broad substrate scope and represents an efficient method for the formation of a C–N bond under mild reaction conditions. Notably, this catalytic methodology provides the regio-isomer of the anthelmintic drug, Thiabendazole, in a lab-scale showing its applicability in the efficient synthesis of such *N*-heterocyclic molecules at industrial levels.

## 1. Introduction

Benzimidazole or 1*H*-1,3-benzodiazole-based heterocycles are structurally similar to naturally occurring nucleotides, i.e., adenine base of the DNA, as well as a component of vitamin B_12_, and have extensively been used in drug synthesis and medicinal chemistry (Figure 1) and display a wide range of biological and clinical applications [1,2,3,4,5,6]. Benzimidazoles can easily interact with the biopolymers of the living systems, which are responsible for their numerous biological activities and functions. In particular, benzimidazole derivatives exhibit antimicrobial [7,8,9], antiviral [10,11], anticancer [12,13,14], anti-inflammatory [15,16,17], and antioxidant [18] activities, whereas various derivatives have been developed as therapeutic agents, such as proton pump inhibitors [2], level modulators [19], and antidiabetics (Figure 1) [20,21].

In general, two main traditional synthetic routes produce benzimidazole derivatives. The first route involves the coupling between *o*-phenylenediamine with carboxylic acids or their derivatives (nitriles, amides, esters, chlorides) [22,23,24]. The second route involves condensation reactions between *o*-phenylenediamine and aldehydes or alcohols via a dehydrogenated coupling, followed by oxidative cyclodehydrogenation [24,25], but in many of these methods, a stoichiometric amount of oxidizing agents is a prerequisite (Scheme 1) [26,27,28,29]. Other methods such as thermal- or acid-promoted synthesis as well as microwave, sonicator, or ultrasound methods are also known [24]. Interestingly, the direct regioselective C-2 arylation of imidazole with aryl halides typically requires the use of a Pd(II)/Cu(I) catalytic system in a large excess of reagents in the presence of additives, high temperature, or pressure [30].

To date, the emphasis is given to developing new, efficient, and “green” methods for the conversion of *o*-phenylenediamine and aldehydes to benzimidazole derivatives. Several heterogeneous catalytic systems that involve inorganic salts [31,32,33,34], zeolites [35,36,37], heterogeneous ionic liquid gel [38], micelles [39], and metal oxides [40,41,42,43,44,45,46,47,48,49] have been reported (Appendix A). This transformation can be also achieved via photocatalytic processes in the presence of photosensitized molecules or semiconductor materials (Appendix A) [50,51,52,53,54,55,56]. In recent years, the synthesis of supported metal nanoparticles as well as their applications, especially in catalysis, is of great interest for further study [57,58,59,60]. Among them, supported gold nanoparticles (AuNPs) have received considerable attention as powerful heterogeneous catalysts in several organic reactions. Interestingly, AuNPs have a small size which provides distinct physical properties and high reactivity [61], and overcome drawbacks of homogeneous gold catalysts. In specifics, AuNPs are not sensitive to air or moisture, promote the reactions under mild conditions, and exhibit high selectivity even in complicated reactions [62]. Several inorganic and organic materials have been used as heterogeneous supports which can disperse and stabilize AuNPs, provide active sites at the metal-support boundary, and influence the oxidation state of gold, thus allowing easy handling and the possibility of recovery [63]. These features have made AuNPs of great importance in the synthesis of heterocycles [64,65,66,67]. 

So far, Ruiz et al. reported the one-pot synthesis of 2-aryl-benzimidazoles from alcohols and diamines in the presence of Au/CeO_2_ nanoparticles in trifluorotoluene at 90 °C (Scheme 2) [68]. The Au/CeO_2_ has been reported as an excellent catalyst for the one-pot synthesis of benzimidazoyl quinoxaline derivatives from the cyclo-condensation of glyceraldehyde and *o*-phenylenediamine derivatives, followed by oxidative coupling with a different *o*-phenylenediamine derivative (Scheme 2) [69]. Tang et al. described the synthesis of 2-aryl-benzoxazoles and benzimidazoles from 2-nitrophenol or 2-nitroaniline, respectively, and alcohols via two hydrogen-transfer processes catalyzed by Au/TiO_2_ in toluene or water at 130–150 °C [70]. Of note is that only the synthesis of 2-phenyl-1*H*-benzo[*d*]imidazole was described using Au/TiO_2_ in toluene or water at 150 °C under N_2_ atmosphere (Scheme 2). Additionally, AuNPs/SBA material was successfully applied for the synthesis of a series of 2-aryl-2,3-dihydroquinazoli-4(*1H*)-ones, as a heterogeneous catalyst in the condensation reaction between 2-aminobenzamide and aromatic aldehydes (result not shown) [71]. In all cases, high temperatures and prolonged reaction time were necessary for reaction completion. Given the importance of this type of transformation and in terms of sustainability, the use of ambient and more eco-friendly heterogeneous conditions for the synthesis of 2-aryl and 2-alkyl benzimidazole derivatives continues to be a long-standing goal of chemical research. In light of our ongoing research directions on developing sustainable catalytic processes to construct *N*-heterocyclic organic molecules of high biological interest [72,73,74], and metal nanoparticle-catalyzed transfer hydrogenation processes for conversion of nitroarenes into amines [75,76,77,78,79,80], herein we report the synthesis of a library of 2-aryl and 2-alkyl benzimidazoles using Au/TiO_2_ as a catalyst at ambient conditions (Scheme 2).

## 2. Materials and Methods

### 2.1. Materials

For the catalytic reactions, we employed commercially available supported gold nanoparticles Au/TiO_2_, Au/Al_2_O_3_, and Al/ZnO, as well as the oxides TiO_2_ (Degussa P25 (Degussa Evonik GmbH, Essen, Germany, anatase/rutile = 3.6/1, BET: 50 m^2^g^−1^, nonporous), and TiO_2_ UV-100 (Hombikat, 100% anatase, BET: 300 m^2^g^−1^), Al_2_O_3_, and SiO_2_. The commercially available catalysts Au/TiO_2_, Au/Al_2_O_3_, and Au/ZnO feature a ca. 1 wt.% Au loading and exhibit an average AuNP size of about 2–3 nm. All commercial heterogeneous catalysts and the salts HAuCl_4_, AuCl, AgNO_3_, and Cu(ClO_4_)_2_·6H_2_O were used without further purification. Catalysts with 1, 2, 3, and 5 wt.% Au loading on anatase mesoporous titania Au/MTA(*x*), *x* = 1, 2, 3, and 5, were synthesized according to our previous works, where the deposition−precipitation method was used [76,77,78]. The products consist of a continuous network of tightly interconnected gold and anatase TiO_2_ nanoparticles and exhibit a large internal surface area (approximately 104−115 m^2^ g^−1^) and a narrow pore size distribution (approximately 7.3−7.5 nm), according to transmission electron microscopy (TEM), to the small-angle X-ray scattering (SAXS), and N_2_ physisorption measurements. Details on synthesis and characterization of the catalysts were described in reference [78]. The average Au particle size measured to be from 4.4 to 5.2 and from 7.1 nm to 9.4 nm, while the gold loading was increased from 1% to 2% and from 3% to 5%, respectively. Product analysis was conducted by ^1^H NMR and ^13^C NMR spectroscopy (Agilent AM 500 and Agilent AM 600, Agilent Technologies, Santa Clara, CA, USA). The identification of the products was realized by comparing the NMR spectra with those of the commercially available pure substances. LC-MS 2010 EV Instrument (Shimadzu, Tokyo, Japan) under Electrospray Ionization (ESI) conditions was used for the determination of the mass spectra. The reagents and solvents were purchased from Sigma-Aldrich (Merck KGaA, Darmstadt, Germany), and Fluorochem (Glossop, UK) and used without further purification. Thin-layer chromatography was performed on Millipore precoated silica gel plates (0.20 mm thick, particle size 25 μm). Chemical shifts for ^1^H NMR were reported as δ values and coupling constants were measured in hertz (Hz). The following abbreviations were used for spin multiplicity: s = singlet, brs = broad singlet, d = doublet, t = triplet, q = quartet, quin = quintet, dd = double of doublets, ddd = double doublet of doublets, and m = multiplet. Infusion experiments were carried out on an Agilent Q-TOF Mass Spectrometer, G6540B model with Dual AJS ESI-MS (Santa Clara, CA, USA). All the compounds (dissolved in LC-MS grade methanol) were introduced into the ESI source of the MS with a single injection of 15 μL of the sample and with a flow rate of 300 μL/min of 100% methanol as a solvent in the binary pump. The experiments were run using a Dual AJS ESI source, operating in a positive ionization mode. Source operating conditions were 330 °C Gas Temp, 8 L/min Gas Flow, Sheath Gas Temp 250 °C, Sheath Gas Flow 10 L/min, and 150 V Fragmentor. Data-dependent MS/MS analysis was performed in parallel with the MS analysis in a centroid mode, using different collision energies (10, 20, 30, 40 V). All accurate mass measurements of the [M+H] ions, or the corresponding major ions in some cases, were carried out by scanning from 100 to 500 m/z. The Q-TOF was calibrated 1 h prior to the infusion experiments by using a calibration mixture.

### 2.2. Catalytic Reaction

The appropriate supported gold catalyst (1 mol % Au, 60 mg of the solid material) was placed in a 5 mL glass reactor (vial), followed by the addition of solvent (3 mL) or solvent mixture CHCl_3_:MeOH (3:1, 3 mL), o-phenylenediamine (0.3 mmol), and the aldehyde (0.3 mmol). The reaction mixture was then stirred at 25 °C (the temperature was maintained using a water bath) for 2 h. The reaction was monitored by thin-layer chromatography (TLC) and after completion, the slurry was centrifuged to separate the solid catalyst from the reaction mixture and washed two times with 3 mL of ethanol. The filtrate was evaporated under vacuum to afford the corresponding products in pure form unless mentioned otherwise. Products, where needed, were purified by column chromatography on a silica gel using a gradient mixture of EtOAc-DCM to afford the desired 2-substituted benzimidazoles in good yields.

### 2.3. Recycling Reaction

Gold catalyst Au/TiO_2_ (1 mol % Au, 40 mg of the solid material) was placed in a 5 mL glass reactor (vial), followed by the addition of solvent mixture CHCl_3_:MeOH (3:1, 1.5 mL), o-phenylenediamine (0.2 mmol), and aromatic aldehyde (0.2 mmol); the reaction mixture was then stirred at 25 °C (the temperature was maintained using water bath), for 2 h. The reaction was monitored by thin-layer chromatography (TLC), and after completion, the slurry was centrifuged to separate the solid catalyst from the reaction mixture and washed three times with the aid of methanol (ca. 3 mL). The filtrate was removed, the solvent was evaporated under vacuum, and the isolated product was determined by ^1^H NMR without any chromatographic purification. The solid material was dried with the use of an oven at 100 °C for 12 h and used without any further purification for the next catalytic run.

## 3. Results and Discussion

### 3.1. Evaluation of the Catalytic Conditions

To optimize the reaction conditions, *o*-phenylenediamine (**1**) and 4-methylbenzaldehyde (**2**) were selected as model substrates. As a starting point, the initial control experiments, in the absence of a catalyst and in 1 mL of MeOH as solvent, showed the formation of the imine **3**, as well as the formation of the desired 2-(4-methylphenyl)-1*H*-1,3-benzodiazole (**4**) and the 1-(4-methylbenzyl)-2-(4-methylphenyl)-1*H*-1,3-benzodiazole (**5**) in comparable yields, even at prolonged reaction time or in the presence of molecular sieves (Table 1, entries 1–4). Recently, a study on the synthesis of benzimidazoles with the use of methanolic solution and short reaction time was reported [81], contradicting the present observations. Additionally, in ethanol, the desired product **4** was formed even after 48 h in 70% yield, accompanied with 30% of the starting aldehyde **2**, as measured by ^1^H NMR of the crude reaction mixture (Table 1, Entries 5 and 6). When the reaction took place in CH_3_CN, similar results were measured with those observed in the case of the methanolic solution described above (Table 1, entries 7–10). In other solvents, such as 1,2-dichloroethane (1,2-DCE), ethyl acetate (EtOAc), and toluene, the corresponding imine **3** was formed as the only product (Table 1, entries 11, 13 and 15), while in CHCl_3_ and tetrahydrofuran (THF), a mixture of the products **3**, **4**, and **5** was observed (Table 1, entries 12 and 14). Even using a mixture of CHCl_3_:MeOH in different ratios, 1:1 and 3:1, no selective formation of the desired **4** was observed even after 18 h (Table 1, entries 16–18). The data summarized in Table 1, shows no selective synthesis of the desired imidazole **4** under the present conditions even at prolonged reaction time. Thus, we continue the development of the catalytic conditions using the above polar and non-polar solvents.

Based on these preliminarily results at the control experiments for the synthesis of the desired product **4**, the development of the catalytic condition was further performed mainly in MeOH and CH_3_CN (Appendix A). Thus, we initially studied the model reaction under homogeneous conditions in the presence of several salts (20 mol %), such as HAuCl_4_ and AuCl, AgNO_3_, and Cu(ClO_4_)_2_**∙**6H_2_O in acetonitrile and methanol (1 mL) at room temperature within 18 h. An equimolar amount of **1** and **2** (0.1 mmol) was dissolved in the appropriate solvent in a 4 mL vial with an open Teflon cap, and the reaction mixture was stirred vigorously. After filtration of the reaction mixture over a short path of silica gel to withhold the catalyst amount and the solvent evaporation, the residue was dissolved in deuterium solvent (ca. CDCl_3_) and the reaction process was determined by ^1^H NMR of the corresponding organic residue. In all cases, mixture of the desired product **4** accompanied by significant amounts of the **5** and the imine **3** were observed (results are shown in Appendix A). 

Moreover, we studied the model reaction in the presence of heterogeneous surfaces such as SiO_2_, Al_2_O_3_, and TiO_2_ (Appendix A, entries 1–7). We have observed that the acidic nature of these surfaces may influence the course of the reaction. When alumina was used as a catalyst in acetonitrile at 2 h, the formation of imine intermediate **3** was observed in 44% yield accompanied by the desired product **4** in 38% yield, as measured by ^1^H NMR (Appendix A, entry 2). Increasing the amount of alumina or the reaction time to 48 h, we noticed a small increase in the yield of **4** (46%) as the imine **3** decreased significantly (18%) (Appendix A, entry 3); while in methanol, the products **4** and **5** were formed in 40% and 60% yields, respectively (Appendix A, entry 4). At the same content, titanium oxide led to the formation of the above products **4** and **5**, in the range of 59% to 39% in MeOH and CH_3_CN, respectively (Appendix A, entries 5–7). Based on literature work [50,51,52,53], where visible light with the presence of a semiconductor or a photosensitizer promoted the synthesis of benzimidazoles, herein the same reaction process was also tried under photochemical conditions, using a Xenon lamp (300 W, *λ* > 300 nm), although no significant changes were observed for the yield of the desired product **4** (Appendix A, entry 8).

Next, we switched to a series of different gold-supported catalysts, i.e., the commercially available Au/TiO_2_, Au/Al_2_O_3_, and Au/ZnO nanoparticles (1% *w*/*w* Au) containing nanoparticles with the size ranging between 2–3 nm. In these cases, 0.1 mmol of the starting compounds **1** and **2** were used with 20 mg of the catalyst (1 mol %) in 1mL of solvent and at 25 °C. It is worth noting that a remarkable increase in the yield of the desired imidazole **4** was observed when methanol was used as a solvent (Table 2, entries 1–3), however, in the presence of Au/TiO_2_, quantitative formation of the **4** was observed within 18 h (Table 2, entry 3). In addition, a series of polar and non-polar solvents were studied herein, i.e., CH_3_CN, 1,2-DCE, EtOAc, and THF (Table 2, entries 4–10), however, no selective formation of the imidazole **4** was observed. Interestingly, in CHCl_3_, the **4** was formed in 94% yield within only 2 h (Table 2, entry 12), compared to the corresponding reactions in EtOH and MeOH, where **4** was observed in 51% and 37% yields accompanied by a significant amount of the imine **3** (Table 2, entries 10 and 11). The same reaction process was also tried under photochemical conditions, using a white-LED apparatus (10.5 W, λ > 380 nm), although, a mixture of the product **3**, **4**, and **5** was observed by ^1^H NMR spectroscopy (Table 2, entry 13). Finally, a mixture of CHCl_3_:MeOH in 3:1 ratio was used, while maintaining a constant temperature of 25 ^°^C, in the presence of the above heterogeneous catalysts (Table 2, entries 14–16). Under these conditions, product **4** was quantitatively formed using Au/TiO_2_ and within only 2 h (Table 2, entry 14). In contrast, Au/Al_2_O_3_ and Au/ZnO led to the formation of **4** in 75% and 58% yield, accompanied by a significant amount of the imine **3** in 25% and 42% yield, respectively (Table 2, entries 15 and 16). In addition, increasing the Au/TiO_2_ amount from 20 mg to 60 mg did not affect the reaction selectivity (Table 2, entry 17). It is worth noting that the catalyst Au/MTA(1) and Au/MTA(2) (4.4 and 5.1 nm of AuNPs) shows high activity and good selectivity for the synthesis of **4** with 91% and 85% yield, respectively (Table 2, entries 18 and 19). In contrast, using the Au/MTA(3) and Au/MTA(5) catalysts with higher nanoparticle size (7.1 and 9.4 nm), a mixture of **3**, **4**, and **5** was observed, with the desired **4** predominating in 54% and 44% yield, respectively (Table 2, entries 20 and 21). In addition, it is noted that the classical concept of nanoparticle concentration/size can play a role in the catalytic procedure [82]. These results indicate a decrease in the selectivity of the imidazole **4** synthesis while the size of the nanoparticle increases. Additionally, gold nanoparticles with small size (<5 nm) are promoted in the present reaction (Table 2, entries 18–21), whereas the support probably significantly influences the reaction process; with TiO_2_ (anatase) to be the most suitable rather than the mesoporous titania MTA, or Al_2_O_3_ and ZnO (Table 2, entries 1–3 and 14–16). This observation supports previous works on the selective reduction of nitroarenes into anilines, where supported AuNPs on TiO_2_, with nanoparticles in the range of 2–5 nm, were the most active catalysts [77,78]. Additionally, a theoretical study on gold particle activity based on the formation of σ-holes, binding sites for Lewis bases, e.g., CO or H_2_O, reported that the catalyst activity increases as the metal nanoparticle size decreases [83]. Similar behavior was observed using Au/Al_2_O_3_ as the catalyst; however, reaction completion toward the synthesis of the desired benzimidazole **4**, along with the best catalytic activity, was observed in prolonged reaction time, 18 h, as shown in Appendix A.

### 3.2. Synthesis of 2-Aryl and 2-Alkyl Benzimidazoles Catalyzed by Gold Nanoparticles

With these optimized conditions, we further explored the scope of this catalytic transformation by incorporating a wide range of commercially available aromatic aldehydes to gain direct access to a library of 2-aryl substituted benzimidazole derivatives in high isolated yields (Scheme 3). For the present heterogeneous conditions, Au/TiO_2_ (60 mg, 1 wt.% or 0.003 mmol Au) was used as the selected catalyst for testing this optimization, as well as 0.3 mmol of the starting aldehyde and 0.3 mmol of the amine **1** were used. Regardless of the electronic nature of the phenyl rings of the aromatic aldehydes, bearing even electron-donating (Me, OMe, OH, OAc, NH_2_, Ph) or electron-withdrawing (Cl, Br, COOMe, COOH, CN, NO_2_) groups, at *o*-, *m*­­- or *p*- position, the desired 2-aryl substituted benzimidazoles (**4**, **6**–**25**) were formed in good to high isolated yields, ranging from 51% to 99% (Scheme 3). In most cases, the desired product was isolated by centrifuging of the crude reaction mixture to separate the solid catalyst from the synthesized 2-aryl benzimidazole. However, in some cases, the product was further purified using a column chromatographic with silica gel and the appropriate mixture of EtOAc/DCM as the eluent (see Appendix A). To expand the application of the present synthetic methodology, 1-naphthaldehyde and heterocyclic aromatic aldehydes, such as the 2-furaldehyde, the 5-(hydroxymethyl)-2-furaldehyde, and the 5-bromo-2-furaldehyde, as well as the thiophene-2-carbaldehyde, give the corresponding products, **26**–**30**, in high yields (80–96%) as shown in Scheme 3. Moreover, the reaction between *o*-phenylenediamine and 6-isopropyl-4-oxo-4*H*-chromene-3-carbaldehyde was performed, and the corresponding substituted benzimidazole **31** was formed in high isolated yield (96%) (Scheme 3). It is worth noting that all the reactions above were also accomplished in the absence of Au/TiO_2_ and under the same conditions. In all cases, mixture of the corresponding imines, the 1,2-disubstituted benzimidazole derivatives, and the desired 2-aryl substituted imidazoles, was observed by ^1^H NMR, without worth-mentioning good selectivity towards the later. These observations support further the importance of the present heterogeneous catalytic system for the synthesis of such *N*-heterocyclic compounds.

With the optimizing conditions in hand, we continued with the synthesis of several 2-alkyl-benzimidazoles, starting with the corresponding aliphatic aldehydes and the diamine **1**. The products **32**–**39** were formed and isolated in 80–98% yields, and the results are summarized in Scheme 4. It is interesting that under the present catalytic methodology, unstable aldehydes such as *trans*-2-phenylcyclopropyl aldehyde and cinnamaldehyde react smoothly, forming the corresponding 2-substituted benzimidazoles **34** and **32**, respectively, in high yields of 88 and 96% (Scheme 4).

These results indicate the broad generality of the present catalytic heterogeneous methodology toward the selective synthesis of 2-aryl and 2-alkyl substituted benzimidazoles. The structures of the products were determined by ^1^H and ^13^C NMR and HRMS (see Appendix A). Exceptions to the generality of the above gold-catalyzed synthetic methodology constitute the reactions of 2-nitrobenzaldehyde **40** and the 6-nitro-4-oxo-4*H*-chromene-3-carbaldehyde **42** with the diamine **1**, where the formation of the corresponding imine intermediate **41** and the precipitation of an unidentified solid residue were observed, respectively (Scheme 5). Consequently, a laboratory-scale procedure was also performed for the direct synthesis of **4** starting from 1 mmol of *o*-phenylenediamine (**1**) and 1 mmol of 4-methylbenzaldehyde (**2**) (Scheme 5). After the completion of the reaction (monitored by TLC), the slurry was centrifuged to separate the solid catalyst from the supernatant. The solid catalysts were washed three times with the aid of methanol ca. 5 mL. The organic layers were collected, and the solvent mixture was evaporated under a vacuum to afford the corresponding product **4** in 96% yield after simple recrystallization over methanol. Based on the present heterogeneous protocol, we also proceeded to the synthesis of the structurally regio-isomer of the antifungal and antiparasitic Thiabendazole, compound **44**, with one-step manner, starting simply from the commercially available aldehyde **43** and diamine **1** (Scheme 5) [84,85,86]. It is interesting that at 25 °C, the only observed product was the corresponding imine in >99% yield. Thus, increasing gradually the reaction temperature to 50 °C and after 18 h, the desired product **44** was observed, purified with column chromatography, and isolated in 92% yield (55 mg) with no byproduct’s formation based on ^1^H NMR analysis. Based on these results, the present one-pot process supports unambiguously the synthetic importance of the proposed catalytic methodology. Previously reported synthetic approaches towards Thiabendazole required external acid additives, harsh reaction conditions, and high temperatures [87,88].

### 3.3. Recycling and Mechanistic Studies

Because of the heterogenicity of the present catalytic system, Au/TiO_2_ can be easily separated from the reaction mixture with a simple centrifuge and the solid catalyst can be washed and reused for the next run. Based on this feature, the recyclability and stability of this material were examined, studying the synthesis of **4** on the above described conditions. In details, 0.2 mmol of **1** and 0.2 mmol of **2** were reacted in the presence of 40 mg of Au/TiO_2_ (1% *w*/*w*) at 1.5 mL of CHCl_3_:MeOH (3:1) and at 25 °C. After the reaction completion within 2 h (monitored by TLC), the slurry was centrifuged to separate the solid catalyst from the supernatant. The solid catalysts were washed three times with the aid of methanol ca. 3 mL and the organic supernatants were collected, and the solvent mixture was evaporated under vacuum to afford the corresponding product **4.** The solid material was dried within the oven at 100 °C overnight and reused for the next catalytic run without any further purification. In Figure 2, the reaction profile based on **4** yield vs. reaction run was presented. As it was shown, the catalyst can be used at least five times without significant loss of its activity towards the synthesis of **4**. This observation is in agreement with previous work on the selective synthesis of benzoxazoles and 2-aryl benzimidazoles, starting with benzyl alcohols and 2-nitroaniline, in toluene and at 150 °C, using Au/TiO_2_ as the catalyst [70]. Additionally, to support the necessity of the AuNPs, as well as to indicate possible leaching of the AuNPs into the reaction mixture, a slurry of 20 mg of Au/TiO_2_ (1% *w*/*w*) in 1 mL of CHCl_3_:MeOH = 3:1 was stirred for 2 h. Then, the supernatant solution was found to be inefficient to promote the selective synthesis of the desired **4**; however, a mixture of **3**, **4**, and **5** was observed by ^1^H NMR, in 22%, 31%, and 47% yields, respectively; similar values to those measured in the corresponding experiment in the absence of a catalyst (Table 1, entry 19). This experiment supports the necessity of the supported AuNPs and implies that Au leaching into the solution is essentially minimal, if any.

Supported AuNPs have been reported as excellent catalysts in cyclo-isomerization processes [64,65,66,67], via the proposed activation of unsaturated bonds [89,90] on the low coordinated electrophilic Au(0) atoms at the corners and edges of nanoparticles [83]. Based on this catalytic process and the present experimental results, a plausible mechanism for the synthesis of the benzimidazole derivatives can be proposed (Scheme 6). At the initial step, the aldehyde and the *o*-phenylenediamine react to generate the imine, which, in the present of the supported gold nanoparticles, led to the intermediate **I** via a cyclization process. This intermediate could undergo protodeauration, which releases the 2-susbtituted-2,3-dihydro-1*H*-benzimidazole intermediate **II** from the chemisorbed species **I**. Then, after dehydrogenation, the corresponding benzimidazole product was formed, also regenerating the catalyst (Scheme 6). We did not detect intermediate **II** by NMR at any stage of the reaction. This notion implies that the dehydrogenation of **II** into benzimidazole occurs faster than the initial cyclization. A similar pathway is proposed in the literature on the synthesis of benzoxazoles and benzimidazoles catalyzed by Au/TiO_2_ [70]. However, an alternative pathway, with the chemisorbed species **I** being transformed, via a dehydrogenation process, into the intermediate **III**, and after protodeauration to the benzimidazole, could also take place. Relevant dehydrogenation pathways are also proposed in the AuNP-catalyzed transformation of 1,2-dihydroquinolines to the corresponding quinolones [91], as well as in the synthesis of pyridines from the in situ generated *N*-propargyl enaminones [92]. Herein, partial charges were used for the intermediates **I** and **III**, based on previous calculations on the bonding nature of chemisorbed intermediates on Au clusters [93,94].

## 4. Conclusions

In conclusion, the commercially available Au/TiO_2_ (2–3 nm AuNPs size) was used as an efficient catalyst for the selective synthesis of 2-aryl and 2-alkyl substituted benzimidazoles. The present heterogeneous catalytic protocol includes the one-step reaction between the corresponding aldehyde and the *o*-phenylenediamine, at ambient conditions, and in CHCl_3_:MeOH (3:1). This reaction has a broad substrate scope and represents a new heterogeneous methodology for practical C-N bond formation under mild conditions, without additional additives and oxidants. The small-size gold nanoparticles (<5 nm) supported on TiO_2_ were found to be the most active species under the present catalytic conditions. The catalyst Au/TiO_2_ could be used at least five times without any significant loss of its catalytic efficacy. The present protocol applied to the lab-scale synthesis of 4-tolylbenzimidazole, as well as to the synthesis of the regio-isomer of the antifungal and antiparasitic Thiabendazole.

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
