# Peer review of "Selective Synthesis of Benzimidazoles from o-Phenylenediamine and Aldehydes Promoted by Supported Gold Nanoparticles"

_nanomaterials, 2020, doi:10.3390/nano10122405_

Round 1

Reviewer 1 Report

The ms describes a new way for the preparative application of supported gold nanopaericles. The preparative goal is significant, the results provide new ways of the synthesis of benzimidazoles. It is a particular merit of the Authors that they provide link to imminent practical application of their results. Some improvement of the literature background and an extensive amendment of the English as well as control of the typing errors appears to be necessary before publication. After these corrections the paper should be published. Some specific remarks:

LITERATURE

The cluster-chemistry background of the supported nanoparticles should be mentioned and the following 2 reviews cited:

Schmid, G. Clusters and Colloids. From Theory to Applications. Wiley-VCH, Weinheim, 2008.

Schmid, G. (Ed.) Nanoparticles. From Theory to Application. Wiley-VCH, Weinheim, 2010.

When discussing the effect of the size of the nanoparticles, it should be mentioned that below a certain limit of size/concentration, the classical concept of concentration could not be applied, the following paper should be cited here:

Maioli, M. & al., Limits of the classical concept of concentration. J. Phys. Chem. B 2016, 120, 7438-7445.

TEXT (Examples of necessary corrections)

Title Benzimidazoles Synthesis => Synthesis of Benzimidazoles

row 12 investigate => investigated; towards => in

row 20 showcasing => showing

row 31 [1], [2],...etc => [1-6] and so also later

row 46 Besides => besides

row 100  TO(Degussa) why not other companies too?

row 129, 173 mL but in row 306 ml  - make uniform notation, mL is better

row 196  ranged => ranging

row 223  ranged => range

row 232  Table 2  - the legend is not very clear

Titles of paragraphs make uniform choice whether capital initial letters or lower case?

The Supplementary files are OK.

Reviewer 2 Report

The papers reports the effective synthesis of the target functional group motif using Au catalysts. In my opinion this paper would be better suited to a method development journal as the goal is driven to achieve 100% yield not determine the rates of the catalysed reaction or information about the nano materials.

Key things to address

Re-use tests are meaningless at 100% yield - it is unclear if i) leached ppm of Au is responsible for the reaction ii) the Au : substrate ratio is so high that even deactivation could occur but not be observed. 

From the blank reactions 70% yield can be achieved using EtOH as solvent without a catalyst (over 48h) with no imine detected suggesting the reaction is occurring homogeneous  - to me it seems like the authors chose a poor solvent system for the catalyst tests to demonstrate the activity - why did the authors not try the catalyst in EtOH as a solvent? 

Can equivalent yields to the catalyst system be achieved in EtOH alone with slightly increased temperature? 

Reviewer 3 Report

Lykakis and co-workers describe a general and efficient protocol for the formation benzimidazols using a heterogeneous and recyclable gold catalyst under mild reaction conditions. The work is suitable for publication in Nanomaterials when the following issues have been addressed:

  • On p 2, the authors write “The second route involves condensation reaction of o-phenylenediamine and aldehydes followed by oxidative cyclodehydrogenation,[24],[25] but in many of these methods, stoichiometric amount of oxidizing agents is a prerequisite (Scheme 1).[26],[27],[28],[29]” However, Scheme 1 also demonstrates alcohols. Please change the wording accordingly.
  • In Table 1, a blank experiment with the same conditions to that of the future standard conditions (Entry 12, Table 2) (reactant concentrations, solvent mixture and reaction time) is missing: please add this to enable comparisons. Please also comment on the reproducibility of the reaction.
  • On p 6 the authors mention particle size and its effect on reactivity, but a discussion the origins of the observed differences is lacking. Do the authors think that there are other aspects apart from the surface area of the Au that matters? Furthermore, Table 2 should contain a column that describes the particle size. In addition, a reaction with a Au/TiO2 catalyst with the same particle size as the commercial catalyst but synthesized in accordance with the method of the catalysts used in entries 17-19 should be performed to assess whether the observed difference is indeed due to particle size or to method of catalyst preparation.
  • Information is missing in Table 2 with respect to the loading of the heterogeneous catalyst (mg) under standard conditions, which makes the comparison of entries 12 and 16 confusing as they both state 1% Au w/w. In the text on p 6 it however appears as if entry 16 has twice the catalyst loading, which would suggest 2% w/w. Please clarify.
  • However, information about the general conditions (reaction time, catalyst loading and reactant concentrations) should be added to Schemes 3 and 4.
  • What does “relative yield” mean (p 11 line 291)?
  • The heading for Section 3.3 should be changed as “recyclic” studies is not a correct term. Overall, the language needs a bit of brushing.
  • It is not clear from the mechanistic scheme on p12 what the actual role of the gold nanoparticle is and the authors should comment on this. Is direct involvement at hand or is the Au simply acting as a Lewis acid?
  • On p12, the authors propose ”proton tropism” as a step in the catalytic cycle. This is not a standard term and requires a definition.
  • Finally, there are many NMR spectra in SI that display additional peaks to those of the product of interest. Please go through the purity of the compounds once more and provide new cleaner spectra along with updated yields when applicable.

Round 2

Reviewer 3 Report

Lykakis and co-workers have addressed several of the previously raised point. However, a few points still remain:

  • Please also comment on the reproducibility of the reaction under standard conditions.
  • Please clarify the discussion of surface area and particle size on p7 further. In addition, a reaction with a Au/TiO2 catalyst with the same particle size as the commercial catalyst but synthesized in accordance with the method of the catalysts used in entries 17-19 should be performed to assess whether the observed difference is indeed due to particle size or to method of catalyst preparation.
  • The heading for Section 2.3 should be changed as “recyclic” studies is not a correct term. Overall, the language needs a bit of brushing.
  • Please clarify the role of gold further in the mechanistic discussion. Although the authors have added new text as well as a reference it is still not clear what they suggest the role of Au to be: directly involved or simply acting as a Lewis acid? Scheme 6 suggests the latter, but it is not clearly stated. Furthermore, references to the statement “The proposed cyclization pathway is different to that occurring under homogeneous ionic Au(I) or Au(III)-catalysis” should be added as well as a description of what mechanism has been proposed in these cases.
  • Please add the data referred to on p 19 to SI: “In addition, increasing the Au/TiO2 amount from 20 mg to 60 mg did not affect the reaction selectivity at 2h (result not shown).”

Author Response

Sincerely yours,

Ioannis Lykakis

Round 3

Reviewer 3 Report

  • The authors have still not provided reproducibility data for the reaction under standard conditions (Table 1, Entry 19). In addition, the provided data for MeOH and MeCN as solvent suggest quite poor reproducibility, which is troubling as this undermines the rest of the conclusions that can be drawn from the screenings. This issue needs to be addressed.
  • With the same loading of a metal catalyst, smaller particles typically result in a larger exposed surface area that substrates can interact with. In this light, it is not surprising that lower activity is observed with larger particle size but this is not clear from the current discussion of the results in Table 2, entries 17-21. However, the results don't only indicate difference in reactivity but also selectivity, which should be commented on as well.

Author Response

Sincerely yours,

Ioannis Lykakis
